# A Wealth Distribution Agent Model Based on a Few Universal Assumptions

**DOI:** 10.3390/e25081236

**Published:** 2023-08-19

**Authors:** Matheus Calvelli, Evaldo M. F. Curado

**Affiliations:** 1Centro Brasileiro de Pesquisas Físicas, Rio de Janeiro 22290-180, RJ, Brazil; matheus.calvelli@gmail.com; 2National Institute of Science and Technology for Complex Systems, Rua Xavier Sigaud 150, Rio de Janeiro 22290-180, RJ, Brazil

**Keywords:** agent-based, economy, information, taxation, complex systems, income distributions

## Abstract

We propose a new agent-based model for studying wealth distribution. We show that a model that links wealth to information (interaction and trade among agents) and to trade advantage is able to qualitatively reproduce real wealth distributions, as well as their evolution over time and equilibrium distributions. These distributions are shown in four scenarios, with two different taxation schemes where, in each scenario, only one of the taxation schemes is applied. In general, the evolving end state is one of extreme wealth concentration, which can be counteracted with an appropriate wealth-based tax. Taxation on annual income alone cannot prevent the evolution towards extreme wealth concentration.

## 1. Introduction

The study of wealth distributions dates back to the late 19th century, when Pareto studied the distribution of land in Italy—an equivalent proxy, at the time, to wealth—and found that, for higher incomes, it was distributed according to a power law, with an exponent α. Later, he found that this distribution was also applicable to Europe as a whole, with an average value α≃3/2, according to Pareto estimates [1]. This new discovery led many economists to believe that this was a robust and stable phenomenon and that, with sufficient data, the same behavior would be found in most other countries in the world, as indeed it was. Clearly, if this behavior is valid in general, it must be based on very general trade behaviors common to all countries in the world, regardless of geographic region, culture, religion, etc.

Trying to have insight into some of these basic behaviors in trade that are common to all countries and that are important for a better understanding of the socioeconomic origins of these distributions, such as those obtained by Pareto, is one of the contributions of this paper. We will discuss some of these common behaviors in trade and their influence on this type of distribution.

The probability density function, P(x), associated with Pareto distribution can be written, in general form, as
(1)$$P\left(x\right)=\left\{\begin{array}{cc} F(x) & \mathrm{for}x<x_{c},\\ \frac{\lambda }{x^{\alpha +1}} & \mathrm{for}x\geq x_{c}\mathrm{where}\lambda >0\mathrm{is}a\mathrm{parameter} \end{array}\right.$$
where for $x<x_{c}$ (*x* can be land, money, etc., in general, any kind of wealth), we have a function F(x), but for $x>x_{c}$, a power law appears, having a typical exponent α. The Pareto distribution, P(x), is usually associated with the cumulative distribution function for the higher values of *x*, i.e.,
(2)$$P\left(x\right)=\int_{x}^{\infty }dx^{\prime }P\left(x^{\prime }\right)\;.$$
Since then, the nature of this distribution of wealth has changed drastically. In particular, the 20th century saw the creation of a strong middle class in some countries ([2,3]), as the industrial revolution, war, and hyperinflation changed the economy of many countries [4]. Interestingly, however, the Pareto distribution would still better describe the tail, while the Boltzmann–Gibbs distribution would best describe the broader part of society (poor and middle class), as can be seen in the income distributions for the US and UK in Figure 1. Therefore, it is not surprising that the ability of the Pareto distribution to describe the wealthier parts of society leads economists to further reinforce their past ideas about its robustness and stability.

Nevertheless, as time goes by and the economic turmoil of the 20th century ends, inequality began to grow at an alarming pace [5], with some economists even predicting a return to 19th-century levels of inequality [5] in some countries, something that many economists thought impossible, further putting into question the stability of the distribution. This increasingly stimulates the theoretical study of the nature of this distribution, trying to better understand its basic causes.

It is important to note that among the various attempts to quantify a little more the measure of inequality, the so-called Lorenz curve, proposed by Max O. Lorenz in 1905 [6], is one of the most important and is used to calculate the Gini index, introduced by Corrado Gini, in 1912 [7], which is currently the standard measure of inequality. Essentially, the Lorenz curve is a graphical representation of inequality, be it wealth inequality or annual income inequality. In this representation, we plot on the abscissa the fraction of the population according to the annual income or wealth and on the vertical axis the fraction of the accumulated wealth or annual income. The Gini index is defined as twice the area contained between the line corresponding to complete equality, i.e., the curve connecting the origin to the point (1,1), and the Lorenz curve.

Among those that studied the inequality from a theoretical point of view, however, agent-based models seem to lead to better results, closer to reality. For example, in [8,9], Chatterjee, A., and Chakrabarti, B. K., considered a simple gaslike model and developed it with increasing levels of complexity. In it, they explore how different types of exchanges can affect a system where, an important point, money is always conserved in any trade interaction ($m_{i}\left(t\right)+m_{j}\left(t\right)=m_{i}\left(t+1\right)+m_{j}\left(t+1\right)$); debt cannot occur; and transactions, where a Δm fraction of money is exchanged, happen randomly between agents. Their model leads, as expected, to a steady state, which is a Gibbs state. Sequentially, the authors added a uniform saving parameter that models the system’s propensity to save, and hence, the amount exchanged has changed. This in turn leads the distribution of wealth to a gamma distribution, thus showing how an additional constraint can change the shape of the distributions. Then, the authors allowed the saving propensity to be distributed among each agent by ρ(λ) and showed that, regardless of the shape of ρ(λ), the asymptotic form of the distribution was always a Pareto one. This result matches the work conducted by Chakraborti and Patriarca [10], which shows that a system composed of subsystems with different degrees of freedom (the propensity to save) leads to a Pareto power law.

Another important work was conducted by Braunstein, Macri, and Iglesias in [11], where they show how, in a complex network, there is a strong link between wealth and the connectivity of agents.

However, when we take a step back from the simulations and look at the data, we can clearly see from the works of Dragulescu in [12] and Banerjee and Yakovenko in [3] and many others ([2,13,14,15,16,17,18,19,20,21,22,23,24,25,26,27,28]) how fundamental and widespread these distributions are, and hence, we expect that this behavior must arise from very basic and common commercial features and interactions.

Therefore, in order to analyze the influence of these universal aspects on trading in distributions, we propose here an agent-based model that is able to bridge these gaps between real-world modeling and fundamental understanding. With this aim, we propose some very basic fundamental (universal) assumptions that are common to any commercial exchange, trying to understand their effects on the distribution of wealth. One of these assumptions is to consider a direct link between information (we further explain this interpretation in the next chapters), represented as the number of connections an agent makes, and wealth. Another assumption, also very general, is the slight trade advantage that a wealthier agent generally has over a poorer agent in any given trade event. Finally, we consider two different types of taxation, a taxation on wealth and a taxation on annual income. In this paper, we will only consider in each scenario (by scenario, we are considering the time evolution of wealth distribution satisfying a set of rules on the connectivity of agents and the trade advantage they have depending on the wealth of the agents, and also the application of only one of two types of taxation, on wealth or on annual income) a single taxation scheme.

We then show how each of these assumptions contributes to the distribution of wealth over time and how they help us understand how different taxation schemes can affect these distributions.

## 2. Outline of the Model

Our goal with this work is to construct a simple agent-based model, rooted in very basic assumptions of business relationships. To do this, we have built our model step by step, introducing complexity along the way, while always maintaining its basic features. Hence, the description of the model, as well as its results, is presented in the same manner, as a step-by-step model built from its simplest form to its most complex, in order to help us understand what the effect of each part is and why it matters. We also separate the description of each version of the model from its results in order to discuss the impact of each parameter and to understand why we have chosen some appropriate numerical values for some of these parameters.

### 2.1. Fundamental Characteristics

We consider a collection of *N* agents, each starting with a given value $w_{0}$ of a continuous variable $w_{i}$, representing the wealth of the agents. The evolution is probabilistic, where, in each Monte Carlo step, each agent is chosen once and trades with other agents, chosen randomly. As the system evolves, agents interact and trade according to four different scenarios. In these interactions, agents can either gain or lose a net amount of money based on the combined wealth of both agents *i* and *j*,
(3)$$\Delta w_{i,j}=\mu \left(w_{i}w_{j}\right)/\left(w_{i}+w_{j}\right)\;,$$
where μ is a constant ∈[0,1]. This function is chosen because it reflects well the relationship of the difference in buying power among agents: the bigger the difference in buying power of one agent relative to another, the greater is its ability to set the amount of money exchanged. This also makes it impossible for agents to trade more wealth than they currently have, a basic requirement. It is important to note, however, that this function is not special and that any other function with a similar qualitative behavior would also work.

At each interaction, both agents play according to a probability distribution and wealth is not conserved, as it happens in reality. Therefore, possibilities where both agents gain (wealth creation) and where both agents lose (wealth destruction) are possible. Since wealth is not conserved in each trade exchange, the total wealth of the system is normalized at the end of each Monte Carlo step.

#### Basic Assumptions

Here, we will introduce the three very basic assumptions that we have adopted and that we believe are generally valid anywhere in the world. Of course, each of these assumptions will be adequately mathematized.

Our first basic assumption, which we believe should be self-evident, is

 **Assumption 1** (First basic assumption)**.**
*The number of trades an agent makes increases with his wealth.*


These agents will then randomly interact with others according to a given connection function, which we define as $f_{c}\left(w_{i}\right)$, which gives how many connections/interactions each agent can make, based on his wealth. For example, if $f_{c}\left(w_{i}\right)=1$, each agent in turn, regardless of its wealth, will perform one interaction, and therefore one transaction, per Monte Carlo step. Notice that in one Monte Carlo step, a given agent may perform more than $f_{c}\left(w_{i}\right)$ interactions, as it may be chosen by other agents in their time.

Therefore, according to the assumption, the connection function $f_{c}\left(w\right)$ mentioned above must be a monotonically increasing function. We will assume here the simplest one, a linear function. Clearly, any other type of monotonically increasing function could be adopted, but the qualitative behavior of the evolution of the system will not change by reasonable choices of the connection function. Only the way the system evolves will change, but not the patterns of the distribution. This assumption will be used in the second, third, and fourth scenarios presented below.

Our second basic assumption, which is also self-evident, is

 **Assumption 2** (Second basic assumption)**.**
*The probability of winning a trade transaction increases with the difference in wealth between the richest and poorest.*


We will introduce a probability of winning a trade exchange that depends on the difference in wealth between the two agents trading. The richer an agent is relative to each other, the higher the probability of winning (in terms of simulation, this means that two “coins” are tossed, one for each agent; hence, they both can win or lose, and situations where one wins and the other loses are also possible) a trade, as is usually the case in any trade negotiation. Hence, when $w_{i}>w_{j}\to P\left(w_{i}|w_{j}\right)>P\left(w_{j}|w_{i}\right)$. The aim is to model the fact that the richest agent in a given trade is usually the one who takes the least risk. This assumption will be used in the third and fourth scenario.

### 2.2. Taxation

Here, we have our third basic assumption:

 **Assumption 3** (Third basic assumption)**.**
*The tax is a monotonic increasing function of wealth.*


We have essentially two main types of taxation: wealth taxation and income taxation (income or capital gains during 1 year, i.e., one stage).

#### 2.2.1. Taxation on Wealth

After a certain number of Monte Carlo steps, here adopted as five, we have what we call a stage, which we can consider equivalent to 1 year. At the end of each stage, a tax is applied on the amount of wealth the agent has at the end of the period. The standard pattern should be that the tax increases with wealth, so we have our third basic assumption.

We then define as a taxation function, to be applied to the wealth of each agent at the end of a stage, the simplest one, a linear function:(4)$$\mathrm{Tax}\left(w_{i}\right)=\left\{\begin{array}{cc} 0, & \mathrm{if}\;w_{i}\leq w_{o}\\ \gamma \left(w_{i}-w_{0}\right)+\sigma w_{0}, & \mathrm{if}\;0<w_{i}<w^{*}\;\mathrm{and}\;\\ \tau , & \mathrm{if}\;\;\;\;\;\;\;\;w_{i}>w^{*}\;, \end{array}\right.$$
where γ,τ,σ∈[0,1]; γ indicates the growth rate of the tax according to wealth, σ the base tax rate, τ the maximum adopted tax rate, and $w^{*}=\left[\tau +\left(\gamma -\sigma \right)\;w_{0}\right]/\gamma$. It is worth noting that taxation here means taxation to be applied on the wealth each agent has. We will consider in this paper only progressive taxation, as it should be; thus, the parameter γ is considered non-negative. Of course, scenarios with negative values of γ will contribute to concentrate more wealth in the hands of fewer agents. The parameter τ could be relevant and is an important political issue today in many countries. The parameter σ, the initial tax rate, is a parameter that is not important at all, but we keep it here for the sake of completeness.

It is also important to note that there is nothing special about the form of this function, chosen here as a linear function. It could also be another type of increasing function, such as a power law. What matters is its behavior—it grows with wealth and has an upper bound. In fact, we explored the quadratic function, and the visible difference was on how quickly the simulations reached their different patterns.

This taxation is then applied to the share of each agent’s wealth $w_{i}$ above a given minimum $w_{0}$ at the end of a fixed number of Monte Carlo steps (called stages), where $w_{0}$ is each agent’s wealth at the beginning of the simulation.

The total tax charged to the *N* agents at the end of a stage is then
(5)$$C_{T}=\sum_{i=1}^{N}\mathrm{Tax}\left(w_{i}\right)\;w_{i}\;\;.$$
This total tax collected at the end of a stage is then redistributed equally among all agents (different scenarios in which, for example, the redistribution favors the poor are also possible and certainly lead to different results and conclusions). Thus, if we let $C_{T}=\mathrm{total}\;\mathrm{tax}\;\mathrm{collected}$, then $C_{i,T}=C_{T}/N$ is the amount of tax that is returned to each agent at the end of a stage.

In the fourth scenario (see Section 2.7 and Section 3.4), we will consider taxation on annual income, but the same third assumption applies: taxation increases with income earned during the previous year.

#### 2.2.2. Taxation on Income or Capital Gain

We have discussed in Section 2.2.1 how to tax the wealth of agents at the end of each year (stage) rather than taxing income or capital gains, which is the much more common form of tax. Therefore, instead of taxing all agents’ wealth $w_{i}$ above the minimum $w_{0}$, we now tax an agent’s annual capital gain whenever it is above the minimum gain ξ. Hence, regardless of how much wealth an agent has, if he has enough capital gain over a year, i.e., a stage, above this minimum (ξ), the agent will be taxed on that amount, in a monotonically increasing way. Then, we can define
(6)$$\mathrm{Capital}\;\mathrm{Gain}\equiv G_{i}=w_{i,t}-w_{i,t-1}\;,$$
where *t* indexes the *t*-th year, i.e., the *t*-th stage—as defined in (Section 2.3). Hence, taxation to be applied to the capital gain of each agent can now be assumed as
(7)$$\mathrm{Tax}\left(G_{i}\right)=\left\{\begin{array}{cc} 0, & \mathrm{if}\;G_{i}\leq \xi \\ \gamma (G_{i}-\xi ), & \mathrm{if}\;\xi <G_{i}<G^{*}\;\mathrm{and}\;\\ \tau , & \mathrm{if}\;\;\;\;G_{i}>G^{*}\;, \end{array}\right.$$
where $\xi \in [0,w_{0}]$ (therefore, it is never greater than the starting point of the systems) and $G^{*}=\left(\tau +\gamma \xi \right)/\gamma$; γ is the the growth rate of the tax according to capital gain.

Notice, however, that here, at the end of a year (a stage), we tax anyone with a capital gain above ξ, regardless of their current (wealth) condition. Therefore, a poor agent ($w_{i}<1$) who has capital gains above the minimum ($G_{i}>\xi$) at a given time *t* will be taxed, even though he is poor. Note, however, that the order of magnitude of taxes in this case (earnings in a year) is very different from the case of a wealth tax. It is important to note that there are many types of taxes that can be collected during the year. There are consumption taxes, which are regressive, affecting poor agents more than rich ones, and taxes on annual income, which are progressive, hurting the rich more. All types of taxes collected during an agent’s year are called here the agent’s annual income tax. By annual here, we mean the earning received during a stage, of course.

Therefore, while in the wealth tax model an agent with $w_{i}>>w_{0}$ can be taxed heavily, since the tax is applied over the agent’s total wealth above $w_{0}$, it also allows poorer agents to build wealth, since they are not taxed until their wealth is at least $w_{i}=w_{0}$. Here, in the case of income tax, the opposite may be true. No matter how poor an agent is, whenever he has a good year, he will be taxed, thus making it difficult for him to build wealth. Meanwhile, extremely wealthy agents could pay almost nothing—relative to their wealth—if their capital gain is not important.

As with taxation on wealth, at the end of the stage (year), the total tax collected that year, which is $\sum_{i}\mathrm{Tax}\left(G_{i}\right)G_{i}$, is redistributed equally among all agents.

Note, however, that in each scenario analyzed in this paper, only one of these two types of taxation is applied, either on wealth or on annual income.

### 2.3. Simulation Setup

We initiate every agent with $w_{i}=w_{0}$ and define a Monte Carlo step when each agent (*N*) in the system finishes his turn, which means


*The agent i, with income $w_{i}$ and number of connections $f_{c}\left(w_{i}\right)=k$, performs all k interactions in one step. These k interactions are randomly chosen.*


The system has no distance (every agent can interact with $f_{c}\left(w_{i}\right)$ other agents, chosen at random). Therefore, since the interacting agents are randomly selected, a given agent *i* may perform more than $f_{c}\left(w_{i}\right)$ interactions per step, since other agents may in turn randomly choose agent i.

We then define that **five Monte Carlo steps** constitute a **stage**, and every step is synchronous: the state of an agent (increase/decrease in wealth) is only updated when the Monte Carlo step is completed (all agents have been updated). Tax collection and redistribution occur only once at the end of the stage. Therefore, Monte Carlo steps can be interpreted as the passage of months, while a stage as the passage of an entire year (annual tax).

We also separate the population in two groups:Agents with $w_{i}\geq 1$, which are shown in the distributions;Agents with $w_{i}<1$, which are taken as the poverty rate and only appear as a percentage.

In the following chapters, we explore different simulation settings (interaction rules, probability, and connection functions) and discuss some of the properties of the model. Note, however, that the functions we have chosen have nothing special about them—we particularly choose the simplest functions whenever possible—it is just their qualitative behaviors that matter. In fact, in the beginning, we tested alternative functions, and the resulting patterns remained unchanged (the speed of evolution may, as mentioned earlier, change depending on the functions chosen).

### 2.4. First Scenario: Raw Model and Taxation on Wealth

This is the simplest, unbiased scenario. Consider a system with the trade rules defined at the beginning of Section 2, with an equal probability of winning a commercial exchange; i.e., the probability that agent *i* will win a commercial exchange with agent *j* is
(8)$$P\left(w_{i}|w_{j}\right)=\frac{1}{2}\;.$$
We also consider the connection function—which, as it is defined in Section 2.1, means how many interactions/transactions an agent will choose at each step—as
(9)$$f_{c}\left(w_{i}\right)=1\;,$$
for any value of $w_{i}$, implying that at each step, each agent chooses only one other agent to trade with.

The numerical simulation results for this model can be seen in Section 3.1.

### 2.5. Second Scenario: The Wealth–Connection Model with Wealth Taxation

In this scenario, we go one step further. We present a simple connection function that links the wealth of an agent with his number of connections,
(10)$$f_{c}\left(w_{i}\right)=\left\{\begin{array}{cc} \alpha \frac{\left(w_{i}-w_{0}\right)}{w_{0}}+1 & \;\;\;\;\;\;\;\mathrm{if}\;\;\;\;\;\;w_{i}\geq w_{0}\\ 1 & \;\;\;\;\;\;\;\mathrm{if}\;\;\;\;\;\;w_{i}<w_{0}\;, \end{array}\right.$$
where α∈[0,1]. The probability of an agent *i* winning a commercial exchange with an agent *j* is still given by Equation (Equation 8).

According to the Function (Equation 10), note that agents will always make at least one interaction and that $f_{c}\left(w_{i}\right)$ is continuous. In order to reproduce this, agents with $f_{c}\left(w_{i}\right)\in R$ have an equivalent probability of having an extra interaction at each Monte Carlo step. For example, an agent *i* with $f_{c}\left(w_{i}\right)=3.14$ will have three connections plus an extra connection with a probability of 14%. A random number will be drawn, and if it is below 0.14, the agent will obtain an extra connection, while if it is above, the agent will only obtain three connections in this round. The numerical simulations associated with this scenario are shown in Section 3.2.

### 2.6. Third Scenario: Favoring the Rich on Transactions and Wealth Taxation

Now, according to our second basic assumption, we introduce a higher probability of winning a commercial transaction for the agent with greater wealth. Until now, each agent had an equal probability of winning a commercial exchange, but now the probability of an agent *i* winning a transaction with an agent *j* will be given by the asymmetric function
(11)$$P\left(w_{i}|w_{j}\right)=\frac{2+\exp(\beta \delta w_{i,j})}{5+\exp(\beta \delta w_{i,j})}\;,$$
where $\delta w_{i,j}=w_{i}-w_{j}$ and β∈[0,1]. This function aims to model the economic bargaining power of an agent. The greater the difference between the wealths of agent *i* and agent *j*, the greater the chance that agent *i* will make a favorable transaction, modeling the fact that the richer agent takes less risk in a trade transaction. At each step, a random number is drawn, and an agent plays this probability with each of the other agents he trades with. Wealth, as always, is not necessarily conserved: if both agents win, wealth is created (both agents earn Δw, Equation (Equation 3)); if only one wins, wealth is conserved (one agent loses Δw, while the other wins); and if both lose, wealth is destroyed. In Figure 2, we can see the behavior of this probability function as a function of β. Notice how the decrease in the probability of winning for the poorer agent is small, while the increase for the rich is significant. The function is not symmetric. This is because if the commercial negotiation is too unfavorable for an agent, he simply does not make the trade (except in very exceptional cases, which are not considered here).

Here, once again, the chosen function is not special, and any other function with similar behavior would work. What matters is the advantage that the rich agent has.

### 2.7. Fourth Scenario: Favoring the Rich in Transactions and Taxation on Annual Income (Capital Gains)

In this scenario, we consider a connection function given by Equation (Equation 10), a probability to win a commercial exchange given by Equation (Equation 11), and a tax on income earned during a year given by Equation (Equation 7).

## 3. Results

Essentially, μ,β, and α only control how fast the system evolves and, therefore, how quickly it goes through the different stages. Higher rates of any of these variables will mean that some of the intermediate distributions will inevitably be skipped because the system will evolve too fast. Therefore, these parameters will be kept constant in all our simulations since we are more interested in following the different stages of evolution (after some tests we adopt from now on the values μ=0.1, β=0.01, and α=1). Similarly, $\sigma w_{0}$, our base tax rate, will be kept at 5% of $w_{0}$ ($\sigma w_{0}=0.05$). The parameters $w_{0}$ and ξ are simply scale parameters, and therefore, their values do not affect the results. Therefore, they are also kept fixed ($w_{0}=10$ and ξ=0) during all numerical simulations.

On the other hand, however, γ and τ could completely change both the evolution of the system and the possible equilibrium states. Therefore, our analysis will consist of varying essentially these two parameters, keeping all others constant.

### 3.1. Raw Model

As defined in Section 2.4, the raw model describes a system without any assumptions that privilege any of the agents. The richest and the poorest have equal opportunities. Therefore, its probability function (bargaining power) and connection function are given by Equations (Equation 8) and (Equation 9). All results presented in this section are for *N* = 100,000 (number of agents) averaged over 100 samples. Taxation is over wealth, given by Equation (Equation 4).

#### Statistics

In Figure 3, we can see that even in a system without any kind of privilege, where no agent has any kind of advantage over another agent, with equitable redistribution of the wealth collected with taxes among agents, there is still some inequality, with the Gini coefficient reaching 0.32. In Figure 3a, we can see that the 99 quantile (quantiles are equivalent to percentiles, and they represent the point that separates the top x% of the population from the rest; for example, the 90 quantile (or q90) separates the top 10% of the population from the other 90%) has a value not greater than $2.5\;w_{0}$, which is not very unequal. In Figure 3b, it can be seen that the 10% richest agents own about 20% of the total wealth, which seems quite reasonable.

However, even in the context of this raw model, where a perfectly egalitarian system still creates inequality, it is easy to infer that in any normal circumstance, where agents do not have perfectly equal opportunities and taxation is not applied to an agent’s total wealth, inequality will likely continue to increase.

The big problem for any society is to avoid great inequalities in order to avoid serious social problems, not necessarily to eliminate them.

### 3.2. Wealth–Trade Link

As defined in Section 2.5, the wealth–connection model describes a system in which the richer the agent, the greater his number of connections (trade exchanges). Therefore, since his bargaining power remains at 50% at all commercial exchanges, due to fluctuations, the wealthier agents end up having higher profits than the poorer agents. The functions that define this scenario are given by Equations (Equation 8) and (Equation 10). All results presented in this section are for *N* = 100,000 averaged over 100 samples. Taxation is on wealth, given by Equation (Equation 4).

#### Statistics

In Figure 4, we can see that the link between wealth and connections allows for greater inequality. Whereas, before, the 99th quantile stabilized around $2.4\;w_{0}$, it now stabilizes at $3.1\;w_{0}$, a 30% increase. Similar differences can also be seen for other statistics. The Gini index went from 0.31 to 0.37, an increase of 20%. The total wealth of the richest 10% went from 20% to 24%, an increase of 20%, and so on.

Therefore, the small advantage of allowing the agent with more wealth to have more connections at each step, which is a fact, and even if his chance of winning a trade exchange is 1/2; i.e., without any advantage, the chances of the agent increasing his wealth increase, leading to a growth of inequality in the population.

### 3.3. Favoring the Rich on Transactions

As defined in Section 2.6, this model describes a system where an agent’s bargaining power, $P(w_{i}|w_{j})$, given by Equation (Equation 11), and his number of connections, $f_{c}\left(w_{i}\right)$, given by Equation (Equation 10), are linked to his wealth. The taxation here is on wealth, given by Equation (Equation 4). Therefore, as an agent becomes richer, his risk decreases both by his increasing bargaining power and by the number of transactions he makes. All results presented in this section are for *N* = 100,000 averaged over 500 simulations.

#### 3.3.1. Distributions

First, we start by exploring the evolution of the wealth distribution in this scenario, which is the main focus of this research: can the model reproduce real-world wealth distribution scenarios with these simple assumptions? The parameter values adopted are τ=0.4 and γ = 1/10,000 (Figure 5 and Figure 6), and γ=1/100 (Figure 7).

On the left-hand side of Figure 5, we can see that at the beginning of the simulation, the system quickly evolves into a Gibbs-type form. However, at stage 5, as the poverty rate (orange line) begins to increase, what resembles a Pareto tail begins to appear. At stage 10, when the poverty rate has passed 0.5% and continues to increase, the Pareto-shaped tail starts to become clearer. At stage 23, its shape reaches exactly the expected behavior, as can be seen in the fitted curve in Figure 6: Gibbs-like middle and poor classes, with a Pareto-shaped tail for the upper 10% of the population and a poverty rate just above 1%. Interestingly, however, as the system reaches that point, the poverty rate begins to decrease due to the wealth tax, as inequality increases. At stage 38, we see that a “secondary” Pareto tail appears with a higher coefficient, much like the distribution for Japanese firms shown by Aoyama et al in [14], showing that, in practice, if given enough time, even the rich begin to differentiate themselves, some much richer than others. Then, at stage 45, poverty continues to decrease, around 0.1% (remember that taxation is levied on wealth), even though inequality is still present and evolving. The much richer, due to the taxation on wealth, rather than annual income, help reduce poverty. The rounded part of the curve for higher values of wealth is due to finite size effects. By increasing the number of agents, this rounded part of the curve shifts to the right. To the right of the distributions, we have a graph of the number of interactions at each stage. We would like to note that, due to the not-so-large number of agents, we cannot claim that these behaviors are true power laws. For this, we would have to run simulations for at least 100 times larger number of agents, which is beyond our scope at the moment. Until the last stage presented in the image (stage 45), the system does not seem to have reached an equilibrium state yet. The tendency towards a condensate state is clear.

Now, when we increase taxation 10 times (γ = 1/10,000 →γ=1/1000), as can be seen in Figure 7, the system still evolves to the expected behavior. However, by reducing inequality, tax revenue is also reduced, and, therefore, the redistribution of wealth. This makes the system apparently reach an equilibrium state (from stage 42 to stage 99) faster and much more egalitarian, but with a much higher poverty rate. This is an important point. It shows that there is a level of taxation above which the system apparently stabilizes, at least for a long time. Now, just as in the previous chapters—where the rules of the system were more egalitarian—inequality still exists. This shows us three things about a system that favors the wealthy (note that, again, this is in the context of a model with perfectly equal tax redistribution; unequal redistribution—those that favor the poor, for example—could lead to different results):The problem of poverty is not simply solved with higher tax rates. How to redistribute the tax collected is also an essential point. Here, the tax has been redistributed equally among agents. A redistribution of tax that favors the poor is likely to decrease the level of poverty. However, this has not been considered in this work, and it would be interesting to analyze this issue in a future work.A strong tax system does not necessarily mean lower poverty rates. As said before, how to redistribute taxes is also a key point.It is not necessary to eliminate inequality in order to end poverty. If the rich are taxed properly—on wealth—and redistribution favors the poor, poverty can be virtually eliminated.

Therefore, these simulations show us a lot about the model. First, it shows that the model is perfectly capable of reproducing, qualitatively, the behavior of wealth distributions in the real world, from more egalitarian societies to strongly unequal societies, where even the richest end up separated into different classes. Second, it shows us that a certain level of inequality generally always exists, but this poverty can be combated with effective taxation on wealth and effective redistribution of these taxes.

#### 3.3.2. Statistics

In order to better analyze the effects of the parameters of the model over its time evolution of the distributions, let us examine some of its statistics. The total tax revenue can be seen in Figure 8 for several values of γ. As the system evolves, lower tax rates lead to higher tax revenues that are applied to fewer and fewer agents. This is because lower tax rates allow a greater concentration of wealth, so fewer and fewer people can reach the minimum wealth required to pay taxes ($w_{0}=10$). This becomes clearer when we look at Figure 9 and Figure 10, which show the evolution of the top 10% and top 1% of the population, respectively. We can see that the 90 quantile initially grows to 2 times $w_{0}$ and then suddenly falls around stage 40 for lower tax rates, although the percentage of wealth held by the top 10% continues to increase. This means that wealth is concentrated in a group of agents (much) smaller than the 10%. This quantifies the effects we saw in the last section: even among the richest agents, a differentiation starts, with some much richer than others (the second Pareto tail we saw). Meanwhile, the simulation with the highest tax rate (γ=0.001) quickly reaches equilibrium (which can be better visualized in Figure 11) and inequality is greatly reduced, although, as we saw in the distributions, poverty is more prevalent. This can be further understood by looking at the Gini coefficient in Figure 11; see how γ=0.0001 (tax rate) leads to lower inequality than a value 10 times higher (γ=0.001). Moreover, notice that in Figure 11, all simulations eventually reach equilibrium, with a stable standard deviation.

### 3.4. Annual Income Tax Model

This scenario is similar to the model described in Section 3.3, where an agent’s bargaining power ($P(w_{i}|w_{j})$) and number of connections are linked ($f_{c}\left(w_{i}\right)$) to his wealth. The difference in this section is how taxation works. Here, instead of taxing the total wealth of an agent at the end of a stage, we tax the amount the agent earned at the last stage, which we call capital gain or annual income. This is the last scenario we are considering, as it is the closest representation to the taxation most commonly used around the world. Therefore, as a connection function, we use Equation (Equation 10); as bargaining power, we use Equation (Equation 11); and as taxation, Equation (Equation 7) is used. All results presented in this section are for N= 100,000 averaged over 500 simulations.

#### 3.4.1. Distributions

The evolution of wealth distributions for the scenario with annual income taxation can be seen in Figure 12. In accordance with what we saw in the previous section, the Gibbs and Pareto tail appears again; see Figure 12, stage 30 (for a proper fit, refer to Figure 13). This time, however, we can see an even steeper second Pareto tail at stage 41 (see Figure 14), showing that inequality among the rich (top 0.1%) has increased even more. In addition, we see a reduction in poverty rate between stages 41 and 51, although we have the highest concentration rate so far. However, as the system continues to evolve, not only does inequality seem to keep growing, but so does poverty, with no signs of reaching equilibrium. This is because, since we are taxing only annual earnings, as the ultrarich possess more and more wealth, there is less and less capital gain to be had in interaction with other agents (since the very rich are few), so tax revenues decrease and the welfare state collapses. These numerical results confirm the widespread intuition among many economists that taxation of annual income fails to balance the concentration of income in a country that, with this type of taxation, always tends towards an ever more extreme concentration. Consequently, according to the model, taxation of wealth and not only of annual incomes seems to be an inevitable policy to avoid a growing concentration of income.

Now, analyzing the results of the shift from wealth taxation to taxation on annual earnings, which is the most widely used type of taxation today in most countries, only confirms the most important part of the model—its ability to represent qualitatively real world wealth distributions based on a few very simple assumptions widely spread around the world. Just these few basic assumptions are already sufficient to represent qualitatively the actual wealth distributions. Of course, we can greatly increase the complexity of the model by adding new economic variables, which also increases the number of parameters, all of which have been incorporated in one way or another in the few parameters used in this model, a kind of coarse-grained one. However, we believe that to understand the basic facts behind the unequal distribution of wealth around the world, the advantages that the rich have in trading more and being more likely to win a deal are among the most important and, as we have seem, are already sufficient to reproduce the qualitative (Pareto) aspects of countries’ wealth distributions in general. Wealth concentration, as these scenarios suggest, is an inevitable fact if there are no effective fiscal policies and a redistribution of wealth with priority to low-income agents. This work suggests that the market alone is incapable of properly handling the problem of wealth concentration, which seems to be circumvented only through effective taxation on wealth, at least in the case of equal redistribution of revenues. This is the scenario that the model clearly shows.

#### 3.4.2. Statistics

If we start by looking at the taxes in Figure 15, we already have a good idea of the impact that the upper tax limit (τ) has on the model. We can see that higher tax limits slow down the time evolution of the model at some stages, but that inevitably they all follow the same path. We can also see that due to the new form of taxation, tax revenues are much lower in all stages. The same behavior can be seen in all other statistics: top 10%, Figure 16; top 1%, Figure 17; standard deviation and Gini coefficient, Figure 18. Here, however, the concentration reaches much higher levels, with the richest 1% holding more than 80% of the total wealth of the population in some cases, while Gini coefficients reach more than 0.8.

Moreover, as we saw in the last section, the change among the rich happens in a faster and stronger way. This time, the 90 quantile reaches two times $w_{0}$ much earlier (stage 20 vs. stage 40 in the wealth tax model) than before and drops to much lower values: in scenario 3, the lowest value of the 90 quantile is just above $0.8w_{0}$ (Figure 9), whereas now it is only $0.25w_{0}$ (Figure 16). A similar trend can also be seen at the 99 quantile (Figure 17). This shows us that the wealth of the population is not, in fact, in the hands of the richest 10% or even the richest 1%, but, actually, in the hands of the 0.0% group (top 0.001%, 0.0001%, 0.00001%, etc.).

Therefore, the impact of the highest level of taxation, τ, does not seem to change the qualitative behavior of the time evolution of the stages; it just delays the same pattern, and it is not a pattern-changing parameter. Then, the important aspects seem to be (i) the value of the parameter γ, (ii) the type of taxation (on wealth or on annual income), and (iii) how the total tax revenue is redistributed (to be analyzed in a future work). All other parameters do not change the tendency of wealth concentration, according to our simulations.

## 4. Conclusions

We show that a simple model that links wealth with trade and favors the rich in commercial transactions can qualitatively reproduce the current wealth/income distributions, with their middle and poor classes having a Gibbs-type distribution and a Pareto tail for the richest parts of the population. Moreover, we show that for capital gain/annual income taxation, the equilibrium state provided by this model is of extreme concentration, while for a wealth tax, it is possible, depending on the parameter values, to reach an equilibrium state that is not the one of extreme concentration of wealth.

The model also shows us that, first, simply increasing taxes does not necessarily lead to lower rates of inequality, because high tax rates lead to a much smaller number of tax-paying agents, thus decreasing the tax revenue to be redistributed, as in standard economic theory. Therefore, in general, higher taxes lead to lower rates of inequality, but there is a special combination of the parameters where this may not be the case; see Figure 11.

However, while it is important to note that this conclusion about the relationship between higher taxes and inequality would likely change if redistribution favored the poor, its fundamental features are still highly likely to be correct. Second, it also shows us that the lower risks (greater bargaining power) and greater number of opportunities (connections) associated with wealthy agents are a central part of the inequality equation, so it is not simply a matter of taxing these agents more, but reducing these differences as well. A high concentration scenario does not exist when opportunities are equal and/or markets are regulated.

Furthermore, the model also shows us that this extreme concentration/inequality can be avoided by taxing wealth, even at low rates, allowing an effective welfare state to functionally eradicate poverty while allowing a healthy economic elite to exist.

Additionally, we can also state with a certain level of confidence that, given the model’s ability to qualitatively reproduce current global trends in economic inequality, as well as its ability to provide results consistent with economic theory, we can understand the current situation based on a few basic assumptions. This means that among the main drivers of inequality, we should consider the problem of unequal opportunities and the difference in risks associated with doing business. Richer agents have much more access to business and a much lower risk rate than the rest of society and, therefore, a much higher probability of winning in any scenario analyzed. The temporal evolution of this scenario leads to growing inequality, inevitably. In addition, the model also makes it very clear that inequality appears naturally and must be actively combated; otherwise, the tendency towards extreme concentration is unstoppable. Therefore, given current global trends, with low taxation, no matter how egalitarian a society may be at the beginning, the system will tend to evolve towards an ever greater concentration.

Moving a little further away from the realm of the model, but based on its findings, we could say that inequality is a problem with multiple origins, but which, at its core, is driven by privileged access to trade and therefore lower risk. However, it is only when this privilege is boundlessly associated with commercial transactions, allowing large corporations to buy and sell as they please, that inequality truly spirals out of control. The market, as we have seen, inevitably creates some level of inequality, which in itself is not a problem if that inequality is not so high. However, when we allow agents with great bargaining power unregulated access to the market, the system truly falls apart.

Looking back while building this model, we also considered much more complex situations, such as distance-based trading advantages between agents, which is also a realistic factor, among others. However, all these more complex alternatives, with many more parameters, exhibited similar behaviors and patterns, making it increasingly clear that the fundamental and simpler characteristics chosen for this model were the driving factors in its behavior.

Finally, the analysis of the effects of unequal redistribution, favoring the poorest, is very important, as it will certainly offer new reasonable scenarios, avoiding the extreme concentration of income we currently see. The study of a hybrid taxation system, in which both taxation on wealth and taxation on annual income coexist, is also an aspect to be considered in future studies.

## Figures and Tables

**Figure 1 entropy-25-01236-f001:**
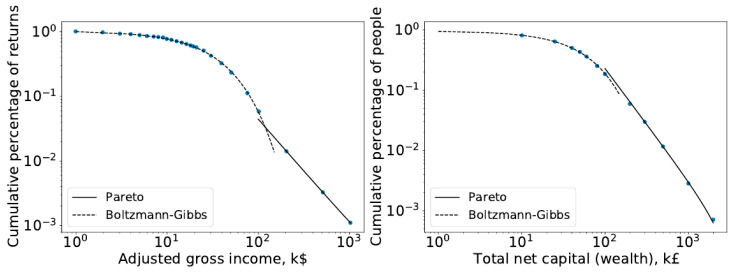
The cumulative probability distribution of net wealth in the US (**left**, 1997) and UK (**right**, 1996) shown in log–log scales. Points represent data from the IRS/HMRC, and solid lines are the fitted lines to the exponential (Boltzmann–Gibbs) and power-law (Pareto) [1].

**Figure 2 entropy-25-01236-f002:**
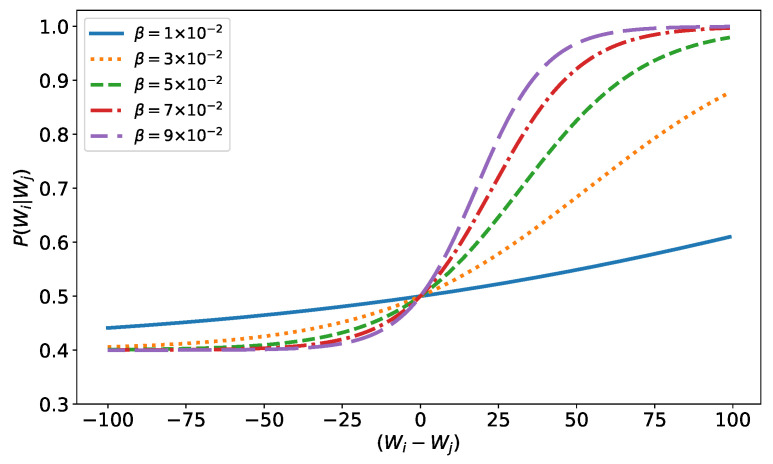
Probability function, Equation (Equation 11).

**Figure 3 entropy-25-01236-f003:**
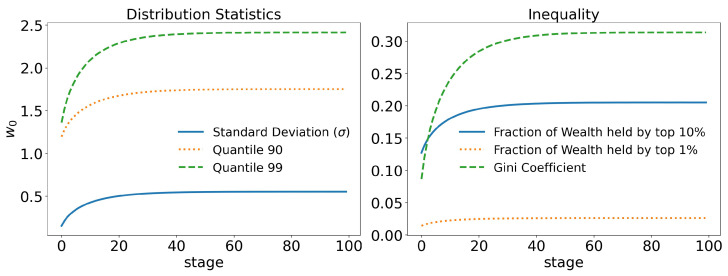
Raw model with taxation on wealth: $\gamma =10^{-3}$ and τ=0.4. In the figure on the left, we can see the average wealth held by the 90 and 99 quantiles, i.e., the 10% and 1% richest agents, respectively, compared with the standard deviation. On the right, the fraction of wealth held by the 10% and 1% richest agents is shown. The time evolution of the Gini index is also shown, stabilizing slightly above 0.3.

**Figure 4 entropy-25-01236-f004:**
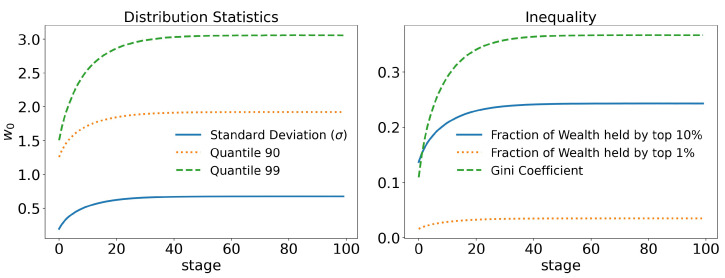
Statistics for the wealth–connection linked model and taxation on wealth: $\gamma =10^{-3}$ and τ=0.4. In the figure on the left, we can see the average wealth held by the 90 and 99 quantiles, i.e., the 10% and 1% richest agents, respectively, compared with the standard deviation. Note that these values are larger than in the raw case, Figure 5. On the right, the fraction of wealth held by the 10% and 1% richest agents is shown. The increase in wealth concentration is evident. Consequently, the Gini index also increases. The time evolution of the Gini index is also shown, stabilizing just below 0.4.

**Figure 5 entropy-25-01236-f005:**
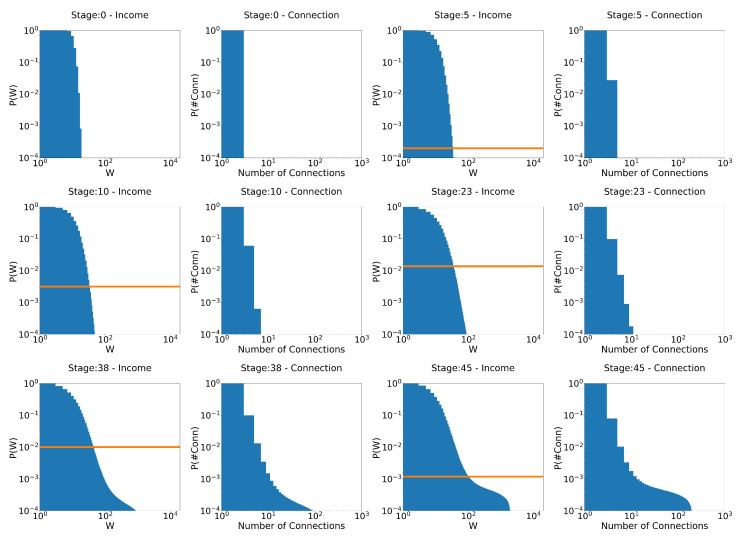
Evolution of the distributions for the model that favors the rich: $\gamma =10^{-4}$ and τ=0.4. At each stage, the figure on the left is the distribution of income, and the figure on the right is the distribution of the number of connections. The orange line is the poverty rate.

**Figure 6 entropy-25-01236-f006:**
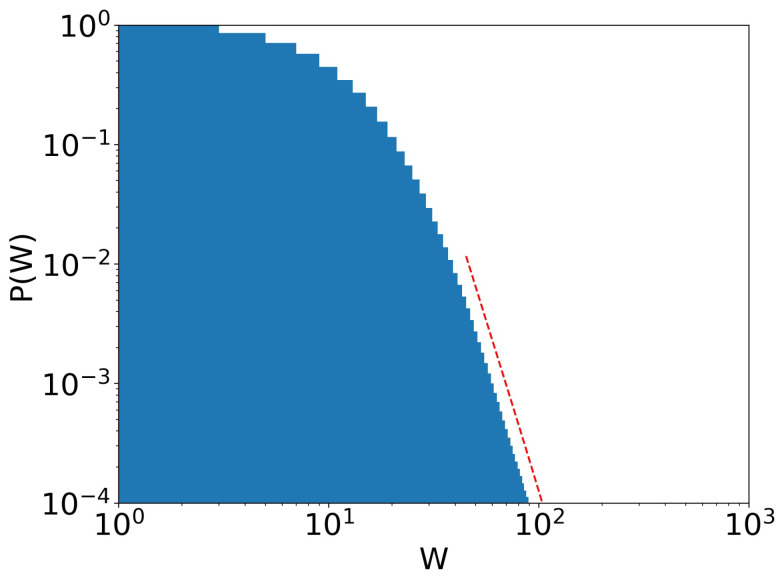
Stage 23 of Figure 5. Pareto tail (dotted red line) is clear, with Pareto exponent α=5.63. $\gamma =10^{-4}$ and τ=0.4.

**Figure 7 entropy-25-01236-f007:**
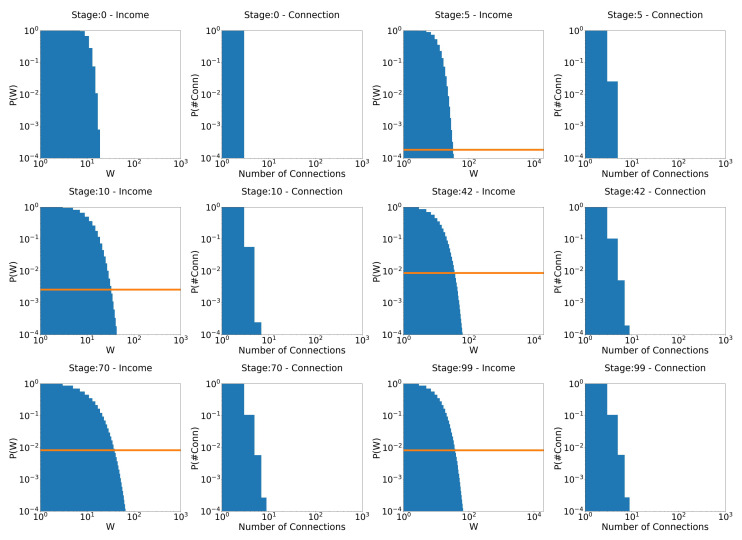
Evolution of distributions for the model that favors the rich: $\gamma =10^{-3}$ and τ=0.4. The orange line is the poverty rate.

**Figure 8 entropy-25-01236-f008:**
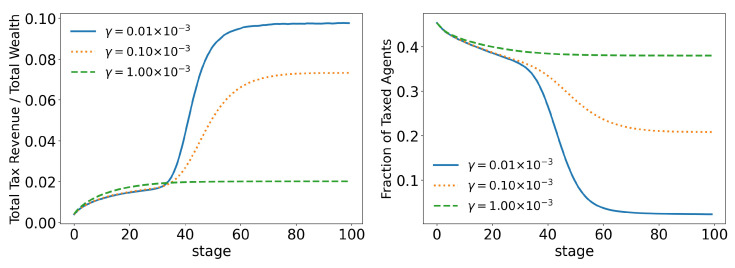
Evolution of total tax revenue and total taxed agents for different values of γ (tax growth rate according to wealth).

**Figure 9 entropy-25-01236-f009:**
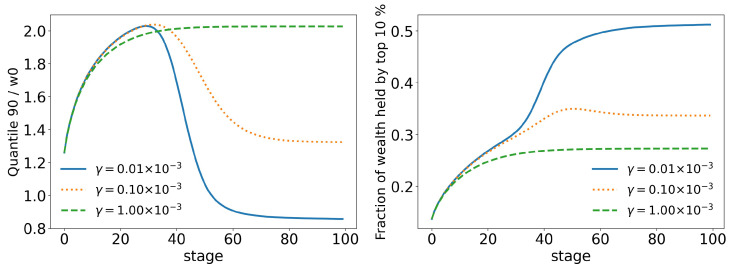
Evolution of the top 10% of agents for different values of γ (tax growth rate according to wealth).

**Figure 10 entropy-25-01236-f010:**
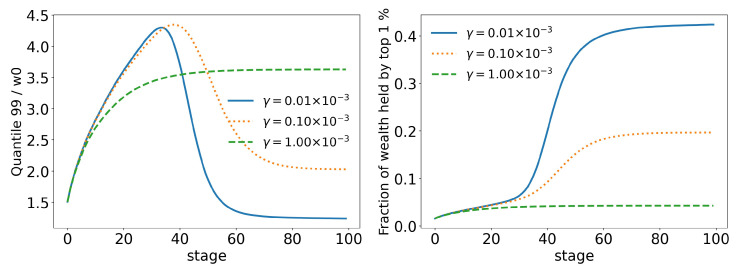
Evolution of the top 1% of agents for different values of γ (tax growth rate according to wealth).

**Figure 11 entropy-25-01236-f011:**
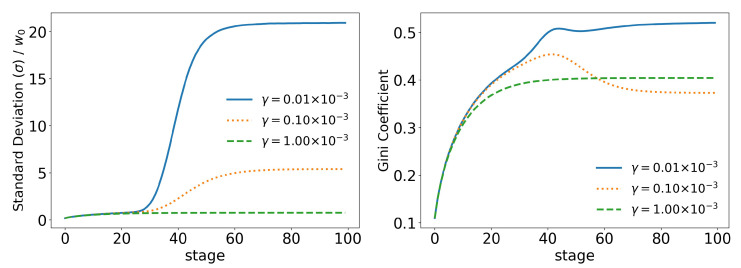
Evolution of the standard deviation (σ) and the Gini coefficient for different values of γ (tax growth rate according to wealth).

**Figure 12 entropy-25-01236-f012:**
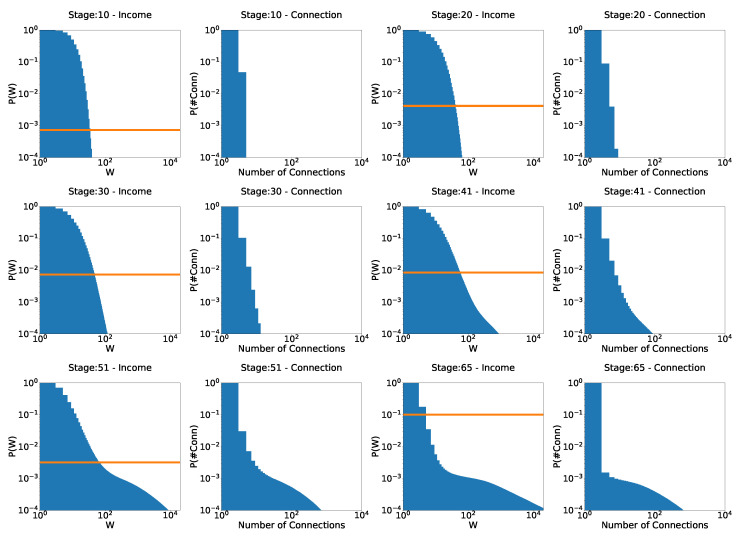
Evolution of probability distributions for the model with capital gain taxation: γ=0.1 and τ=0.4. The orange line is the poverty rate.

**Figure 13 entropy-25-01236-f013:**
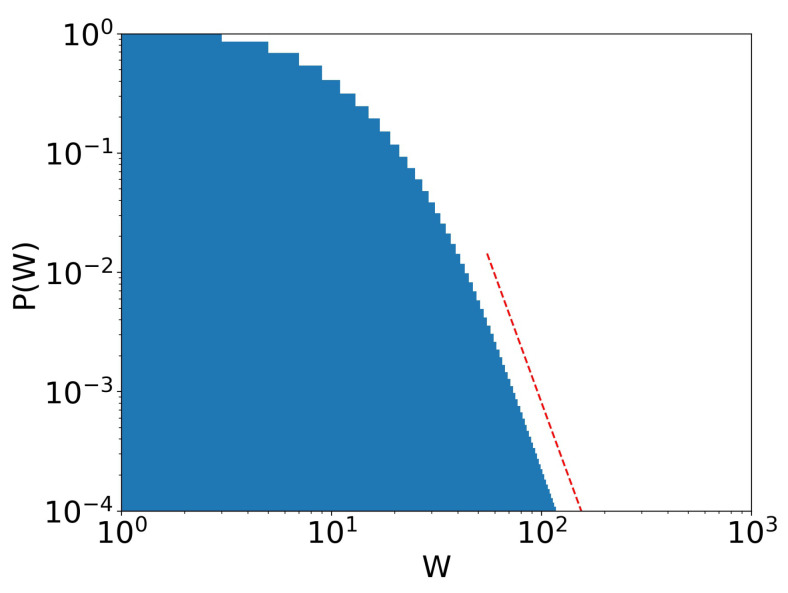
Stage 30 of Figure 12. There is a Pareto tail (dotted red line) with α=4.82.

**Figure 14 entropy-25-01236-f014:**
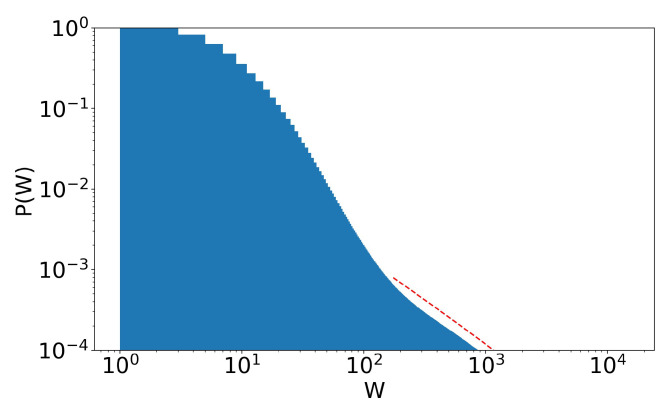
Stage 41 of Figure 12, scenario of annual income taxation. A second Pareto tail appears, with α=1.06 (dotted red line). γ=1/10 and τ=0.4.

**Figure 15 entropy-25-01236-f015:**
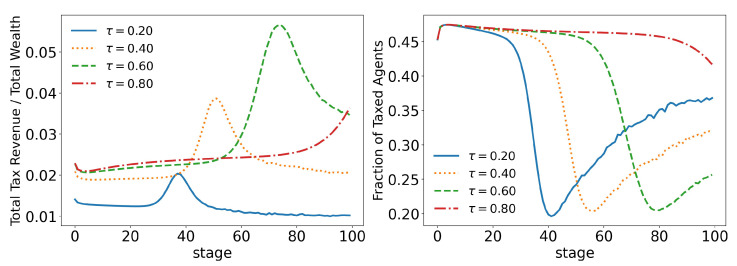
Evolution of total tax revenue and total taxed agents for different values of τ (tax limit). γ=0.1.

**Figure 16 entropy-25-01236-f016:**
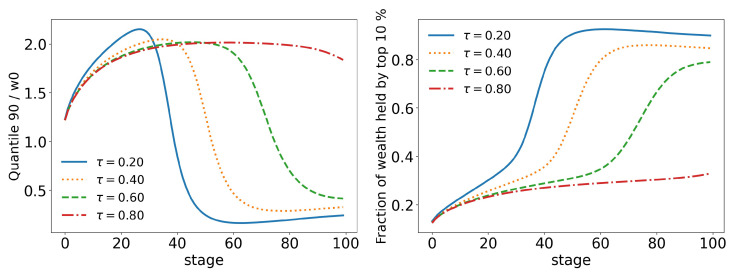
Evolution of the top 10% of agents for different values of τ (tax limit) and γ=0.1.

**Figure 17 entropy-25-01236-f017:**
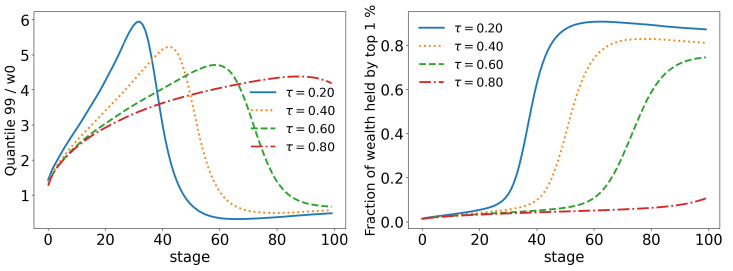
Evolution of the top 1% of agents for different values of τ (tax limit) and γ=0.1.

**Figure 18 entropy-25-01236-f018:**
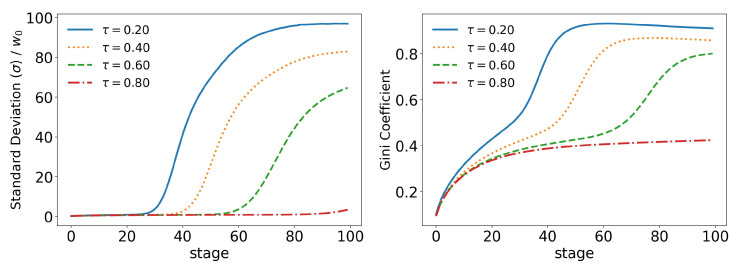
Evolution of the standard deviation (σ) and the Gini coefficient for different values of τ (tax limit) and for γ=0.1.

## Data Availability

Not applicable.

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
