# Peer review of "A Wealth Distribution Agent Model Based on a Few Universal Assumptions"

_entropy, 2023, doi:10.3390/e25081236_

Round 1

Reviewer 1 Report

Please, see my comments and criticisms in the attached pdf file (commneted over the original MS pdf)

Reviewer 2 Report

Given in the attached file.

Round 2

Reviewer 1 Report

The authors have addressed all my criticisms and suggestion, so
in my view the MS can be a good contribution to Entropy journal.

A few final remarks and suggestion that could be useful:

1) 2.2 Taxation

Where it says:

"We have essentially two main types of taxation: total wealth taxation and annual tax-
ation (income or capital gains during one year)."

Why not use just one word for each kind of the taxation:

*wealth taxation* and *income taxation*

So then:

2) 2.2.1 *Taxation on wealth*

and

3) 2.2.2 *Taxation on income*

4) In the expression of w* (after eq(4) in P.7) there are two pair of parenthesis and bracket pair.
The brackets should be in place of the second pair of parenthesis.

5) Expression (6) shows that the taxation is over each operation (with gain over the threshold),
so the word "annual" in this context could be confused regarding the previous taxation system
-which, in fact, is applied after a "year".

6) If they decide to take my advice on this nomenclature, they should check the results (and figure captions)
for coherence.

Author Response

Dear Editor,

We are pleased that our paper, entitled  "A wealth distribution agent model based on a few universal assumptions”, has been accepted. 

Referees 2 and 3 have no further suggestions. Referee 1 made some simple suggestions, which improve some aspects, and which we decided to take into account. His suggestions are mentioned below, as well as our answers.

We hope this new version is suitable for publication in the Special Issue of Entropy.

Yours sincerely,

Evaldo Curado and Matheus Calvelli

Reply to referee one's comments

—————————————————————————————————-

Thanks again to referee 1 for his suggestions

  1. 1) 2.2 Taxation

Where it says: "We have essentially two main types of taxation: total wealth taxation and annual taxation (income or capital gains during one year)." Why not use just one word for each kind of the taxation: *wealth taxation* and *income taxation*
So then:
2) 2.2.1 *Taxation on wealth*

and
3) 2.2.2 *Taxation on income*

Our answer:

According to the referee’s comment, we modified these sentences in sections 2.2, 2.2.1 and 2.2.2. The modifications are highlighted in blue.

2) In the expression of w* (after eq(4) in P.7) there are two pair of parenthesis and bracket pair. The brackets should be in place of the second pair of parenthesis.

Our answer:

Thank you for this remark. The brackets have been put in the correct place. See two lines below eq. (4).

3) Expression (6) shows that the taxation is over each operation (with gain over the threshold), so the word "annual" in this context could be confused regarding the previous taxation system - which, in fact, is applied after a "year".

6) If they decide to take my advice on this nomenclature, they should check the results (and figure captions)
for coherence.

Our answer:

The parameter t in eq. (6) indexes a stage, i.e., a year. This taxation is applied only at the end of a stage. We modified the first sentence below eq. (6) to make it clearer. We also included the word “stage" in the third line below eq. (7) and at the end of the same paragraph we included a new sentence clearer defining the word “annual”. All these changes are highlighted in blue.

Reviewer 2 Report

Accept.

Author Response

Dear Editor,

We are pleased that our paper, entitled "A wealth distribution agent model based on a few universal assumptions”, has been accepted. 

Referees 2 and 3 have no further suggestions. Referee 1 made some simple suggestions, which improve some aspects, and which we decided to take into account. His (her) suggestions are mentioned below, as well as our answers, highlighted in blue.

We hope this new version is suitable for publication in the Special Issue of Entropy.

Yours sincerely,

Evaldo Curado and Matheus Calvelli

Reply to referee one's comments

—————————————————————————————————-

Thanks again to referee 1 for his suggestions

  1. 1) 2.2 Taxation

Where it says: "We have essentially two main types of taxation: total wealth taxation and annual taxation (income or capital gains during one year)." Why not use just one word for each kind of the taxation: *wealth taxation* and *income taxation*
So then:
2) 2.2.1 *Taxation on wealth*

and
3) 2.2.2 *Taxation on income*

Our answer:

According to the referee’s comment, we modified these sentences in sections 2.2, 2.2.1 and 2.2.2. The modifications are highlighted in blue.

2) In the expression of w* (after eq(4) in P.7) there are two pair of parenthesis and bracket pair. The brackets should be in place of the second pair of parenthesis.

Our answer:

Thank you for this remark. The brackets have been put in the correct place. See two lines below eq. (4).

3) Expression (6) shows that the taxation is over each operation (with gain over the threshold), so the word "annual" in this context could be confused regarding the previous taxation system - which, in fact, is applied after a "year".

6) If they decide to take my advice on this nomenclature, they should check the results (and figure captions)
for coherence.

Our answer:

The parameter t in eq. (6) indexes a stage, i.e., a year. This taxation is applied only at the end of a stage. We modified the first sentence below eq. (6) to make it clearer. We also included the word “stage" in the third line below eq. (7) and at the end of the same paragraph we included a new sentence clearer defining the word “annual”. All these changes are highlighted in blue.